# Literature Review of Deep Network Compression

**Ali Alqahtani** [1,2], **Xianghua Xie** [1,*] **and Mark W. Jones** [1]

1   Department of Computer Science, Swansea University, Swansea SA2 8PP, UK; amosfer@kku.edu.sa (A.A.); m.w.jones@swansea.ac.uk (M.W.J.)
2   Department of Computer Science, King Khalid University, Abha 62529, Saudi Arabia
*   Correspondence: x.xie@swansea.ac.uk

**Abstract:** Deep networks often possess a vast number of parameters, and their significant redundancy in parameterization has become a widely-recognized property. This presents significant challenges and restricts many deep learning applications, making the focus on reducing the complexity of models while maintaining their powerful performance. In this paper, we present an overview of popular methods and review recent works on compressing and accelerating deep neural networks. We consider not only pruning methods but also quantization methods, and low-rank factorization methods. This review also intends to clarify these major concepts, and highlights their characteristics, advantages, and shortcomings.

**Keywords:** deep learning; neural networks pruning; model compression

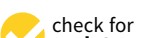



## 1. Introduction

In recent years, deep learning has rapidly grown and begun to show its robust ability in representation learning, achieving remarkable success in diverse applications. This achievement has been possible through its ability to discover, learn, and perform automatic representation by transforming raw data into an abstract representation. The process of deep learning utilizes a hierarchical level of neural networks of different kinds, such as multilayer perceptron (MLP), convolutional neural networks (CNNs), and recurrent neural networks (RNNs). This hierarchical representation allows models to learn features at multiple abstraction levels, meaning that complicated concepts can be learned from simpler ones. Neurons in earlier layers of a network learn low-level features, while neurons in later layers learn more complex concepts [1].

The achievement of neural networks in a variety of applications is accompanied by a dramatic increase in computational costs and memory requirements. Due to the sufficient amount of data and advanced computing power, neural networks have turned into wider and deeper architectures, driving state-of-the-art performances in a wide range of applications. Despite their great success, neural networks have a massive number of parameters, and their significant redundancy in the parameterization has become a widely-recognized property [2]. The over-parametrized and redundant nature of neural networks incur expensive computational costs and high storage requirements. To classify a single image, the VGG-16 model [3], for instance, requires more than 30 billion float point operations (FLOPs), and contains about 138 million parameters with more than 500 MB of storage space. This presents significant challenges and restricts many CNN applications. Recognizing the importance of network units can help to reduce the model complexity by discarding less essential units.

Most of the computational complexity originates in the convolutional layers due to massive multiplication and addition operations, although they contain less parameters due to parameter sharing. The number of FLOPs is utilized as a popular metric to estimate the complexity of CNN models. The FLOPs in convolutional layers are calculated as follows [4]:

$$\text{FLOPs} = 2HW(C_{in}K^2 + 1)C_{out}, \tag{1}$$

where H, W, $C_{out}$ refers to the height, width and number of channels in the output tensor, K is the kernel size, $C_{in}$ denotes the number of input channels, and 1 is the corresponding bias. In contrast, most of the weights parameters exist in fully-connected layers, where the dense vector-matrix multiplications are very substantial resources. Table 1 represents the complexity of several CNNs' architectures, which consist of two parts: (1) the computational complexity is essentially related to the convolutional layers and (2) the parameters in fully-connected layers dominate complexity. Accordingly, reducing the computational complexity of the convolutional layers became the focus of most model acceleration methods, while model compression methods mainly target the parameters of the fully-connected layers.

**Table 1.** Summary of Modern CNNs with their performance, computational, and parameter complexities in ImageNet database. M/B indicates million/billion ($10^6/10^9$), respectively.

| Year | Network | Layers (#) | Size | Performance | | Computational Complexity | | | Parameter Complexity | | |
|------|---------|-----------|------|-----------|-----------|--------|----------|--------|---------|----------|--------|
| | | | | Top-1 (%) | Top-5 (%) | FLOPs | Conv (%) | FC (%) | Par.(#) | Conv (%) | FC (%) |
| 2012 | **AlexNet** [5] | 8 | 240 megabyte | 36.70 | 15.30 | 724 M | 91.9 | 8.1 | 61 M | 3.8 | 96.2 |
| 2014 | **VGGNet** [3] | 16 | 528 megabyte | 23.70 | 6.80 | 15.5 B | 99.2 | 0.8 | 138 M | 10.6 | 89.4 |
| 2014 | **GoogleNet** [6] | 22 | 88 megabyte | 22.10 | 6.30 | 1.6 B | 99.9 | 0.1 | 6.9 M | 85.1 | 14.9 |
| 2015 | **ResNet** [7] | 50 | 98 megabyte | 20.74 | 5.25 | 3.9 B | 100 | 0 | 25.5 M | 100 | 0 |

These complexities present significant challenges and restrict many applications. For instance, deploying sizeable deep learning models to a resource-limited device leads to various constraints as on-device memory is limited [8]. Therefore, reducing computational costs and storage requirements is critical to widen the applicability of deep learning models in a broader range of applications (e.g., mobile devices, autonomous agents, embedded systems, and real-time applications). Reducing the complexity of models while maintaining their powerful performance creates unprecedented opportunities for researchers to tackle major challenges in deploying deep learning systems to a resource-limited device. Network pruning focuses on discarding unnecessary parts of neural networks to reduce the computational costs and memory requirements associated with deep models. Pruning approaches have received considerable attention as a way to tackle over-parameterization and redundancy. Consequently, over-parameterized networks can be efficiently compressed and allow for the acquisition of a small subset of the whole model, representing the reference model with fewer parameters [9]. There is no authoritative guide for choosing the best network architecture; a model may require a certain level of redundancy during model training to guarantee excellent performance [10]. Hence, decreasing the size of a model after training can be an effective solution.

Pruning approaches were conceptualized in the early 1980s and '90s, and can be applied to any part of deep neural networks [11–17]. Optimal Brain Damage (OBD) by LeCun et al. [13], and Optimal Brain Surgeon (OBS) by Hassibi et al. [14] are considered pioneering works of network pruning, demonstrating that several unimportant weights can be removed from a trained network with little accuracy loss. Due to expensive computation costs, these methods are not applicable to today's deep models. Obtaining a sub-network with fewer parameters without reducing accuracy is the main goal of pruning algorithms. The pruned version, a subset of the whole model, can represent the reference model at a smaller size or with a smaller number of parameters. Over-parameterized networks can therefore be efficiently compressed while maintaining the property of better generalization [18].

In this paper, we present an overview of popular methods and review recent works on compressing and accelerating deep neural networks, which have received considerable attention from the deep learning community and have already achieved remarkable

progress. The types of compression methods discussed below are intended to provide an overview of popular techniques used in the research of deep neural network compression and acceleration.

The rest of this paper is organized as follows. Section 2 describes the methodology used to collect related research papers and the scope of the literature. Section 3 presents a detailed review of deep network compression, derived from our general classification for deep network compression and acceleration. Section 4 summarizes and discusses the future challenges reported within our collection. Finally, concluding remarks and summary are provided in Section 5.

## 2. Methodology

### 2.1. Survey Search Methodology

A variety of concepts and methods are involved in obtaining a sub-network with fewer parameters without reducing accuracy. Our search methodology was to collect, study, and analyze many papers in the field of deep network compression and network pruning. In our search of the literature, we started by looking at each individual journal and conference in the computer vision and deep learning communities. We performed a keyword search, e.g., 'network compression', 'network pruning', 'network acceleration','model compression and acceleration', or 'compact network architectures'. We list all the literature sources searched in Table 2.

**Table 2.** A list of literature sources searched for Deep Network Compression. We mainly use IEEE Xplore, the ACM Digital Library, the Elsevier Library, the Springer Library, and Google Scholar to search for literature.

| Conferences and Journals | Papers |
| :---: | :---: |
| Advances in Neural Information Processing Systems | 13 |
| International Conference on Learning Representations | 12 |
| IEEE Conference on Computer Vision and Pattern Recognition | 5 |
| CoRR | 6 |
| International Conference on Machine Learning | 3 |
| European Conference on Computer Vision | 2 |
| International Conference on Acoustics, Speech and Signal Processing | 2 |
| British Machine Vision Conference | 2 |
| Pattern Recognition | 2 |
| IEEE Transactions on Pattern Analysis and Machine Intelligence | 1 |
| IEEE International Conference on Computer Vision | 1 |
| Computer Vision and Image Understanding | 1 |
| International Conference on Pattern Recognition | 1 |
| Nature communications | 1 |
| International Conference on Applications of Intelligent Systems | 1 |
| Signal Processing | 1 |
| IEEE Access | 1 |
| IEEE International Joint Conference on Neural Networks | 1 |
| International Joint Conference on Artificial Intelligence | 1 |
| Total | 57 |

### 2.2. Survey Scope

In scope: To fulfil the scope of our survey, we selected papers that focus on deep network compression and model pruning approaches. We found and collected 57 papers to include in our deep network survey. We pay attention to compression methods and pruning levels for all papers whether a model is pre-trained or trained from scratch.

Out of scope: We restrict our literature to papers that include a review of deep network compression approaches. Papers that focus on data compression are out of our survey's scope. Unlike model compression, data compression (i.e., text compression [19], genomic

compression [20], and image compression [21–23]) forms a central role to handle the bottleneck of data storage, transmission, and processing.

### 2.3. Survey Classification

The recently advanced approaches for deep network compression and acceleration presented in this work can be classified into three categories: pruning methods, quantization methods, and low-rank factorization methods.

## 3. Deep Network Compression

### 3.1. Pruning Methods

This section illustrates approaches that have been proposed to prune non-informative parts from heavy, over-parameterized deep models, including weights (i.e., parameters or connections) and units (i.e., neurons or filters). The core of network pruning is eliminating unimportant, redundant, or unnecessary parts according to the level of importance. Pruning methods can be applied to pre-trained models or trained from scratch and are further categorized into two classes according to pruning level: weights level and units level. Weight-based pruning eliminates unnecessary, low-weight connections between layers of a neural network while unit-based methods remove all weight connections to a specific unit, where both income or outgoing weights are removed.

#### 3.1.1. Weight-Based Methods

Several weight-based methods have been proposed to prune non-informative connections. Recently, Han et al. [24] introduced a pruning method to remove connections whose absolute values are smaller than a predefined threshold value calculated using the standard deviation of a layer's weights. The network is then retrained to account for the drop in accuracy. Although Han's framework received significant attention and has become a typical method of network pruning, it focuses on the magnitude of weights, relies on iterative pruning and fine-tuning, and requires a particular software/hardware accelerator not supported by off-the-shelf libraries. Moreover, the reliance on a predefined threshold is not practical and too inflexible for some applications.

Liu et al. [25] showed the possibility of overriding the retraining phase by random reinitialization before the retraining step, which delivers equal accuracy with comparable training time. Furthermore, Mocanu et al. [26] replaced the fully-connected layers with sparsely-connected layers by applying initial topology based on the Erdős–Rényi random graph. During training, fractions of the smallest weights are iteratively removed and replaced with the new random weights. Applying initial topology allows for the finding of a sparse architecture before training; however, this requires expensive training steps and obviously benefits from iteratively random initialization. The random connectivity of non-structured sparse models can also cause poor cache locality and jumping memory access, which extremely limits the practical acceleration [27].

Through an iterative pruning technique, Frankle et al. [28] found that over- parameterized networks contain small sub-networks (winning tickets) that reach test accuracy comparable to the original network. The obtained sparse network can be trained from scratch using the same initialization as the original model to achieve the same level of accuracy. Their core idea was to find a smaller architecture better suited to the target task at the training phase. In a follow-up study, Frankle et al. [29] found that pruning networks at initialization values does not work well with deeper architectures, and suggested setting the weights to those obtained at a given early epoch in training. Various extensions have been developed for further improvement and to experimentally analyze the existence of the lottery hypothesis in other types of networks [30–33].

To overcome the weaknesses associated with unstructured pruning, strategies corresponding to group-wise sparsity-based network pruning have been explored. Wen et al. [27] proposed the Structured Sparsity Learning (SSL) method, which imposes group-wise sparsity regularization on CNNs, applying the sparsity at different levels of their structure

(filters, channels, and layers) to construct compressed networks. Lebedev et al. [34] also employed group-wise sparsity regularization to shrink individual weights toward zero so they can be effectively ignored. Furthermore, Zhou et al. [35] incorporated sparsity constraints on network weights during the training stage, aiming to build pruned DNNs. Although this proved successful in such sparse solutions, it results in damage to the original network structure and there is still a need to adopt special libraries or use particular sparse matrix multiplication to accelerate the inference speed in real applications.

It can be argued that the use of weight-based methods suffers from certain limitations. The need to remove low-weight connections means that important neurons whose activation does not contribute enough due to low-magnitude income or outgoing connections could be ignored. Moreover, the overall impact of weight-based pruning on network compression is lower than neuron-based methods. Pruning a neuron eliminates entire rows or columns of the weight matrices from both the former and later layers connected to that neuron, while weight-based methods only prune the low-weight connections between layers. To process the resulting sparse weight-matrices, some methods also require a particular software/hardware accelerator that off-the-shelf libraries do not support. Despite these drawbacks, the weight-based methods can be applied in combination with unit-based methods to add extra compression value.

### 3.1.2. Unit-Based Methods (Neurons, Kernels, and Filters)

Unit-based methods represent a pruning approach proposed to eliminate the least important units. He et al. [36] developed a simple unit-based pruning strategy that involves evaluating the importance of a neuron by summing the output weights of each one, and eliminating unimportant nodes based on this. They also apply neuron-based pruning utilizing the entropy of neuron activation. Their entropy function evaluates the activation distribution of each neuron based on a predefined threshold, which is only suitable with a sigmoid activation function. Since this method damages the network's accuracy, additional fine-tuning is required to obtain satisfactory performance. Alqahtani et al. [37] proposed a majority voting technique to compare the activation values among neurons and assign a voting score to quantitatively evaluate their importance, which helps to effectively reduce model complexity by eliminating the less influential neurons. Their method simultaneously identifies the critical neurons and prunes the model during training without involving any pre-training or fine-tuning procedures.

Srinivas et al. [38] also introduced a unit-based pruning method by evaluating the weights similarity of neurons in a layer. A neuron is removed when its weights are similar to that of another in its layer. Mariet et al. [39] introduced Divnet, which selects a subset of diverse neurons and merges similar neurons into one. The subset is selected based on activation patterns by defining a probability measure over subsets of neurons. As with others, these pruning methods require software/hardware accelerators that are unsupported by off-the-shelf libraries and a multi-step procedure to prune neurons.

Filter-level pruning strategies have been widely studied. The aim of these strategies is to evaluate the importance of intermediate units, where pruning is conducted according to the lowest scores. Li et al. [40] suggested such a pruning method based on the absolute weighted sum, and Liu et al. [41] proposed a pruning method based on the mean gradient of feature maps in each layer, which reflects the importance of features extracted by convolutional kernels. Other data-driven pruning methods have been developed to prune non-informative filters. For instance, Polyak et al. [42] designed a statistical pruning method that removes filters based on variance of channels by applying the feature maps activation variance to evaluate the critical filters. Unimportant filters can also be pruned according to the level of importance. Luo's [43] pruning method is based on the entropy of the channels' output to evaluate the importance of their filters, and prunes the lowest output entropy, while Hu et al. [44] evaluated the importance of filters based on the average percentage of zero activations (APoZ) in their output feature maps.

Furthermore, Luo et al. [10] proposed the ThiNet method, which applies a greedy strategy for channel selection. This prunes the target layer by greedily selecting the input channel with the smallest increase in reconstruction error. The least-squares approach is applied to indicate a subset of input channels which have the smallest impact to approximate the output feature map. A general channel pruning approach is also presented by Liu et al. [45], where a layer-grouping algorithm is proposed to find coupled channels automatically. Then a unified metric based on Fisher information is derived to evaluate the importance of a single channel and coupled channels. These methods tend to compress networks by simply adopting straightforward selection criteria based on statistical information. However, dealing with an individual CNN filter requires an intuitive process to determine selective and semantically meaningful criteria for filter selection, where each convolution filter responds to a specific high-level concept associated with different semantic parts. The most recent work is a CNN pruning method inspired by neural network interpretability. Yeom et al. [46] combined the two disconnected research lines of interpretability and model compression by basing a pruning method on layer-wise relevance propagation (LRP) [47], where weights or filters are pruned based on their relevance score. Alqahtani et al. [48] proposed a framework to measure the importance of individual hidden units by computing a measure of relevance to identify the most critical filters, introducing the use of the activation of feature maps to detect valuable information and the essential semantic parts to evaluate the importance of feature maps.

It could be argued that compressing a network via a training process may provide more effective solutions. Ding et al. [49] presented an optimization method that enforces correlation among filters to converge at the same values to create identical filters, of which, redundant ones are safely eliminated during training. He et al. [50] proposed a filter pruning method which prunes convolutional filters in the training phase. After each training epoch, the method measures the importance of filters based on L2 norm, and the least essential filters are set to zero. He et al. [51] later iteratively measured the importance of the filter by calculating the distance between the convolution kernel and the origin or the geometric mean based on which redundant kernels are identified and pruned during training. Liu et al. [52] trained an auxiliary network to predict the weights of the pruned networks and estimate the performance of the remaining filters. Moreover, Zhonghui et al. [53] applied a training objective to compress the model as a task of learning a scaling factor associated with each filter and estimating its importance by evaluating the change in the loss function. AutoPruner [54] embedded the pruning phase into an end-to-end trainable framework. After each activation, an extra layer is added to estimate a similar scaling effect of activation, which is then binarized for pruning. A significant drawback of iterative pruning is the extensive computational cost; and pruning procedures based on training iterations often change the optimization function and even introduce hyper-parameters which make the training more challenging to converge.

### 3.2. Quantization Methods

Network quantization is a deep network compression procedure in which quantization, low precision, or binary representations are used to reduce the number of bits when representing each weight. Typical deep networks apply floating point (e.g., 32-bit) precision for training and inference, which is accompanied by a dramatic increase in computational costs, memory, and storage requirements. Several works [55–57] introduced low bit-width models with a high level of accuracy, considering both activation and weight quantization. In the parameter space, Gong et al. [58], and Wu et al. [8] applied Kmeans clustering on the weight values for quantization. As a result, the network weights are stored in a compressed format after completing the training process, which allows them to reduce storage requirements and computational complexity. 8-bit quantization of the parameters has been proved to achieve significant speedup with minimal accuracy loss [59]. Suyog et al. [60] showed that truncating all parameters to 16-bits can result in a significant reduction in memory usage and floating point operations without compromising accuracy.

Others have proposed to simultaneously prune and quantize the weights' magnitudes of a trained neural network. Han et al. [61] iteratively eliminated the unnecessary weight connections and quantized the weights, which were then encoded to single-bit precision by applying Huffman coding for further compression. This achieved state-of-the-art performance with no drop in model accuracy. Soft weight-sharing [62] was also developed to combine quantization and pruning approaches in one retraining procedure. Chen et al. [63] introduced a HashedNets model that applied a random hash function on the connection weights to force the weights to share identical values, resulting in a reduction in the number of trainable parameters by grouping them into hash buckets. These pruning approaches typically generate connection pruning in CNNs. In advanced cases, 1-bit quantization is used to represent each weight. A number of binary-based methods exist to directly train networks with binary weights (i.e., BinaryNet [64], BinaryConnect [65], and XNOR-Networks [55]), who shared the idea of learning binary weights or activation during the training process.

The disadvantages of binary networks include significant performance drops when dealing with larger CNNs, and they ignore the impact of binarization on accuracy loss. To overcome this, Hou et al. [66] employed a proximal Newton algorithm with a diagonal Hessian approximation to minimize the overall loss associated with binary weights, and Lin et al. [67] quantized the representations at each layer when computing parameter gradients, converting multiplications into binary shifts by enforcing the values of the neurons of power-of-two integers.

### 3.3. Low-Rank Factorization Methods

Low-rank approximation (factorization) is applied to determine the informative parameters, applying matrix or tensor decomposition. A weight matrix is factorized into a product of two smaller matrices, performing a similar function to the original weight matrix. In deep CNNs, the greatest computational cost results from convolution operations, so compressing the convolutional layers would improve overall speedup and compression rate. Convolutional units can be viewed as a 4D tensor, as the fact that the 4D tensor consists of a significant amount of redundancy drives the idea of tensor decomposition, which is an effective way to eliminate redundancy.

Low-rank factorization has been utilized for model compression and acceleration to achieve further speedup and obtain small CNN models. Rigamonti et al. [68] post-processed the learned filters by employing a shared set of separable 1D filters to approximate convolutional filters with low-rank filters, and Denton et al. used low-rank approximation and clustering schemes to reduce the computational complexity of CNNs. Jaderberg et al. [69] suggested using different tensor decomposition schemes, achieving double speedup for a particular convolutional layer with little drop in model accuracy. Low-rank factorization has also been used to exploit low-rankness in fully-connected layers. Denil et al. [9] utilized a low-rank decomposition of the weight matrices which learned from an auto-encoder to reduce the number of dynamic parameters, while Sainath et al. [70] showed that low-rank factorization of the last weighting layer significantly reduces the number of parameters. Lu et al. [71] adopted SVD to composite the fully-connected layer, attempting to design compact multi-task deep learning architectures. Low-rank approximation is made in a layer-by-layer fashion: at each layer, the layer is fine-tuned based on a reconstruction objective, while keeping all other layers fixed. Following this approach, Lebedev et al. [72] applied the non-linear least-squares algorithm, a type of Canonical Polyadic Decomposition (CPD), to approximate the weight tensors of the convolution kernels. Tai et al. [73] introduced a closed-form solution to obtain results of the low-rank decomposition through training constrained CNNs from scratch. The Batch Normalization layer (BN) is utilized to normalize the activations of the latent, hidden layers. This procedure has been shown to be effective in learning the low-rank constrained networks.

Low-rank factorization approaches are computationally expensive because they involve decomposition operations. They also cannot perform global parameter compression

as low-rank approximation is carried out layer-by-layer [74]. Undertaking sufficient re-training is the only technique which can be used to achieve convergence when compared to the original model. Despite their downsides, these approaches can be integrated with conventional pruning methods to obtain more compressed networks for further improvement.

## 4. Discussion of Challenges and Future Directions

After we present a review of network compression and acceleration works and classify them into three categories, we, here, highlight opportunities and potential future directions. Reducing the complexity of models while maintaining their high performance creates unprecedented opportunities for researchers to tackle the challenges of deploying deep learning systems to a resource-limited device and increasing deep network models' applicability to a broader range of applications. Choosing well-suited methods to compress and accelerate deep neural networks relies on the applications and requirements. For instance, pruning and low-rank factorization-based methods may present effective solutions when dealing with pre-trained models. In particular tasks (i.e., object detection), low-rank factorization may be better suited when accelerating convolutional layers, while the pruning method can be adopted when compressing fully-connected layers.

Applying initial topology or random connectivity of sparse models allows for the finding of a sparse architecture. Although this process has been proved successful [25,26,28–33], it is still part of a relatively young and emerging field. Most of the proposed methods damage the original network structure, demonstrating the necessity to adopt some special libraries or to use particular sparse matrix multiplication to accelerate the inference speed in real applications. Random connectivity causes cache and memory access issues so that the acceleration of even high sparsity models is very limited. Therefore, more theoretical analysis requires further study to better understand how to improve sparse models and introduce more effective methods.

The effectiveness of deep representations has been shown to extend to network pruning [46,48]. For instance, the pruning methods presented in [48] make use of quantifying the importance of latent representations, compressing and accelerating CNNs for image classification tasks, including CIFAR object recognition, CUB-200 fine-grained classification, and ImageNet large-scale object classification. Applying such pruning methods to real applications in several different computer vision tasks, including object detection, semantic segmentation, image generation, image retrieval, and style transfer, is a fertile avenue for future research, as these visual tasks require richer knowledge and more abstract feature representation than image classification, meaning that they may face a sharp reduction in model performance [75,76]. Research could visually explore how such applications are capable of making use of our pruning method, particularly semantic segmentation and image generation.

Several filter-level pruning strategies, proposed for CNN compression and acceleration approach, mainly focus on filter-level pruning [4,10,40,41,44,48], where removing the unimportant filter in its entirety does not affect the network structure. This would allow for more significant compression and acceleration by other compression approaches, such as the parameter quantization approach and low-rank factorization methods. Although these approaches are computationally expensive and cannot perform global parameter compression, integrating them with filter-level pruning methods would obtain more compressed networks for further improvement. It would also be fruitful to explore the usage of a hybrid scheme for network compression, where the advantages of each network compression category can be exploited to prune models further.

There are also several challenges and extensions we perceive as useful research directions. The first would be to extend the multi-step filter-level pruning framework and combine it with an iterative pruning method to more deeply explore the problem and accomplish effective CNN compression and acceleration, as pruning a network via a training process may provide more effective solutions. Secondly, most pruning methods are data-driven based, so their speed efficiency is a significant concern. Although pruning-

based methods inspired by neural network interpretability achieved better results, it can be time-consuming to complete their process. Although a few images are selected from each category to form our evaluation set used to find the optimal channel subset, the [48] method still requires more than seven minutes to estimate IoU scores and MV values for one block only on ResNet-50 and ImageNet. Therefore, parallel implementation could be a promising solution, where CNN-based methods are more suitable for efficient parallelization benefit on both CPUs and GPUs. Consideration of a set of nodes, filters, and layers for pruning, instead of one by one in a greedy manner is also worthwhile to study in our future work.

Overall, the potential for deep network compression is vast; the field has many open problems to understand and explore. The remarkable advancement of neural network interpretability should encourage the development of efficient methods for network compression and acceleration to facilitate the deployment of advanced deep networks.

## 5. Conclusions

The over-parametrized and redundant nature of network models incurs expensive computational costs and high storage requirements, presenting significant challenges, and restricts many of their applications. Therefore, reducing the complexity of models while maintaining their powerful performance is always desirable. This paper has discussed necessary background information for deep network compression. We presented a comprehensive, detailed review of recent works on compressing and accelerating deep neural networks. Popular methods such as pruning methods, quantization methods, and low-rank factorization methods were described. We hope this paper can act as a keystone for future research on deep network compression.

**Author Contributions:** Conceptualization, methodology, validation, formal analysis, investigation, writing—original draft preparation, writing—review and editing: All. All authors have read and agreed to the published version of the manuscript.

**Funding:** This work was supported by the Deanship of Scientific Research, King Khalid University of Kingdom of Saudi Arabia under research grant number (RGP1/207/42).

**Institutional Review Board Statement:** Not applicable.

**Informed Consent Statement:** Not applicable.

**Data Availability Statement:** Not applicable.

**Conflicts of Interest:** The authors declare no conflict of interest.

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
