# Peer review of "Literature Review of Deep Network Compression"

_informatics, doi:10.3390/informatics8040077_

Round 1

Reviewer 1 Report

The authors concentrated their efforts on a survey of the literature on Deep Network Compression.

Deep Network Compression is a topic that is now trending and in high demand. The amount of available material is adequate to produce high-quality literature reviews and other reviews.

A literature review is often required to have at least one research topic or problem that is resolved in a given group of acceptable articles, or at the very least refers the reader to further research. Unfortunately, the research topic and the methodology used to create this Literature review are both absent from this document. Because there is no methodological foundation for this overview, it is simply a collection of articles and techniques that have been subjectively chosen and do not add anything new to the knowledge of the topic.

I would suggest the authors add the methodological portion of the study. 

Furthermore, please include a Discussion section in which the authors will outline future research and research gaps that they have identified within LR. Add also information about the constraints of this research, as well as its advantages and disadvantages, as well as probable blunders.

Author Response

                    We thank the reviewer for their comments and feedback that contributed to enhance the content and presentation of our submission. In the revised version, We added a new section (2. Methodology), including more a further discussion and analysis of the literature search methodology,  literature scope, literature classification in response to a different reviewer comment as requested. Table.2 has also been added to show the list of literature sources and the number of papers. 

                    We thank the reviewer again for their comments and feedback that contributed to enhancing the content and presentation of our submission. In the revised version, We added a new section (4. Discussion of Challenges and Future Directions), which adds a further discussion and analysis of this in response to the reviewer comments. We consider the future of the field of Deep Network Compression and highlight potential research directions.  The constraints of Deep Network Compression and their limitations are presented in detail.

Reviewer 2 Report

The paper deals with a literature review of deep network compression.

A good introduction is done.

The paper is good described. I have low suggestions:

  • Perform a sub-section about the general purpose compression by relying on the specific usage (such as genomic, image, until your subject)
  • check any typos error

For additional references about compression I suggest the following citations:

  • M. Amich, P. D. Luca and S. Fiscale, "Accelerated implementation of FQSqueezer novel genomic compression method," 2020 19th International Symposium on Parallel and Distributed Computing (ISPDC), 2020, pp. 158-163, doi: 10.1109/ISPDC51135.2020.00030.
  • Liu, L., Zhang, S., Kuang, Z., Zhou, A., Xue, J. H., Wang, X., ... & Zhang, W. (2021, July). Group Fisher Pruning for Practical Network Compression. In International Conference on Machine Learning (pp. 7021-7032). PMLR.
  • Li, Z., Zhang, Z., Zhao, H., Wang, R., Chen, K., Utiyama, M., & Sumita, E. (2021). Text compression-aided transformer encoding. IEEE Transactions on Pattern Analysis and Machine Intelligence.

Author Response

We thank the reviewer for their comments and feedback. In the revised version, We added a sub-section in (2. Methodology), which is the literature scope. This sub-section has clearify our in scope and out of scope papers. We restrict our literature to surveys that include a review of deep network compression approaches. Papers that focus on data compression are out of our survey’s scope (i.e. text compression, genomic compression, and image compression).

We included the listed paper [2] to its related Section (3.1.2. Unit-based Methods (Neurons, Kernels and Filters)) and paper [1,2] have been added to out of scope section as the papers have looked at data compression methods while our paper focuses on deep network compression.

Round 2

Reviewer 1 Report

Dear authors, thank you for amending the submitted text following the reviewers' suggestions. The added methodological section is adequate to the literature review findings, and the included Discussion part smoothly supplements the research findings. I am pleased to report that the authors have made improvements, and I can now recommend the manuscript for publication.